# Predictors of Moderate or Severe Cognitive Impairment at Six Months of the Hip Fracture in the Surgical Patient over 65 Years of Age

**DOI:** 10.3390/jcm11092608

**Published:** 2022-05-06

**Authors:** Enrique González-Marcos, Enrique González-García, Paula Rodríguez-Fernández, Jerónimo J. González-Bernal, Esteban Sánchez-González, Josefa González-Santos

**Affiliations:** 1RACA 11 Artillery Regiment, Cid Campeador Military Base, 09193 Burgos, Spain; enriquegojs@gmail.com; 2Traumatology and Orthopedic Surgery Service, Burgos University Hospital, 09006 Burgos, Spain; enriqueglezgar@yahoo.es; 3Department of Health Sciences, University of Burgos, 09001 Burgos, Spain; mjgonzalez@ubu.es; 4Department of Medicine, University of Vic, 08500 Barcelona, Spain; estebansg2001@gmail.com

**Keywords:** hip fracture, clinical features, cognitive impairment, surgical patient, elderly

## Abstract

Background: cognitive impairment is known to be very common in patients with hip fractures, but studies are needed to help understand the relationship between both events. Our goal was to determine the relationship between moderate or severe cognitive impairment and hip fractures during the six months following that episode. Methods: a retrospective longitudinal study was conducted on a sample of 665 people over 65 years of age. The main variable of the study was cognitive impairment at six months of fracture, assessed using the Pfeiffer scale (PS). Other data related to clinical features were also collected for further analysis. Results: binary logistic regression analyses showed that the main factors related to moderate or severe cognitive impairment at the sixth month of the fracture were age (OR = 1.078), initial cognitive impairment (OR = 535.762), and discharge (OR = 547.91), cognitive worsening at the sixth month with respect to the time of admission (OR = 7.024), moderate dependence on admission (OR = 15.474) and at six months (OR = 8.088), poor ambulation at discharge (OR = 5.071) and institutionalization prior to admission (OR = 5.349) or during the first semester after fracture (OR = 6.317). Conclusions: this research provides evidence about the clinical factors that predict moderate or severe cognitive decline at the sixth month in patients undergoing surgery for a hip fracture.

## 1. Introduction

Cognitive impairment is a very frequent event in patients with hip fractures, functional deterioration being greater in those people who present a previous cognitive deterioration [1,2] and is a risk factor related to the incidence of this pathology in the elderly [3], especially in women over 65 years of age [4]. Dementia and hip fractures follow a common explanatory pathogenic pattern, involving multi-fall syndrome associated with gait disorders, instability, sarcopenia, as well as bone fragility due to osteoporosis, as described by Friedman et al. in their study [5].

Moderate or severe cognitive impairment is defined by a number of errors in answering the Pfeiffer scale questionnaire ≥5, which implies the loss of the user’s autonomy. People with moderate or severe cognitive impairment, with hip fractures and associated comorbidities on admission (heart disease and arterial hypertension), that have been prescribed some drugs (antiaggregants, anti-dementia drugs, neuroleptics), are complicated by constipation and are more likely to be institutionalized with an inability to walk at discharge. People without prior cognitive impairment are not complicated by “delirium” during admission [6]. Complications with acute episodes called delusions or “delirium” are also common during admission for hip fractures [7], and are defined as a sudden onset of confusion, loss of attention, and impairment of the level of consciousness [6]. In any case, cognitive impairment and “delirium” are associated with older age and negatively influence the function and survival of hospitalized patients with hip fractures [8].

In Castilla y León the prevalence of cognitive impairment is 13.1% in the general population over 65 years of age [9] and 19% in urban areas [10], more frequently in women, diabetics, and people with lower educational levels [10]. A total of 44.1% of the elderly hip-fractured patients in Spain have a cognitive impairment, understood with more than three errors in the Pfeiffer scale (PS) or Short Portable Mental State Questionnaire (SPMSQ) according to the National Registry of Hip Fractures due to Fragility carried out in 2017 [11], and the incidence of hip fracture in Spain in 2008 was 103 cases per 10,000 inhabitants, with an upward trend since 1997 [12].

The number of people with hip fractures and cognitive impairment is very high, and since 1998 it has been known that dementia is a factor linked to dependence and impairment of function according to the Katz Index [13], as well as hip fracture, which translates into a great functional limitation in people suffering from both pathologies. Previous cognitive impairment, alcohol abuse, and age are the main factors related to the appearance of moderate cognitive impairment in the surgical patient, but the lack of established knowledge and the need for new studies that allow the identification, prevention, and early treatment of the causes that lead this type of patient to suffer from delirium are recognized [14,15].

Although the importance of cognitive status in the functional and vital diagnosis of the surgical patient is evident, specific studies are required that include the relationship between cognitive impairment and hip fractures during the months following it [16]. Knowing how and to what extent cognitive impairment and hip fractures are related could contribute to less functional limitations and would provide novel scientific evidence to better manage such comorbidity and design decision support tools, as well as specific approaches and methodologies. Therefore, our study aims to determine the relationship between significant cognitive impairment and the different clinical characteristics of patients aged 65 years or older with hip fractures, during the six months after that episode. The hypothesis of the study is that there will be various factors, after a hip fracture, that will lead to cognitive impairment.

## 2. Materials and Methods

### 2.1. Study Design—Participants

Retrospective longitudinal study, all patients were treated at the University Hospital of Burgos (HUBU).

Inclusion criteria: Patients aged 65 years or older, who by a low energy mechanism suffered a hip fracture, in the biennium 14 March 2019–14 March 2021. All patients admitted to the HUBU with these characteristics were included in the study, followed after discharge from the outpatient clinics of the Orthopedic Surgery and Traumatology Service of the same hospital through face-to-face and non-face-to-face consultations through interviews with the patients, their families, and/or responsible caregivers.

Exclusion criteria: Patients with peri-prosthetic fractures, peri-synthesis fractures, pathological fractures, that is, on bones affected by primary tumor or metastasis, were excluded from the study, likewise, patients who were referred to other hospitals without completing the treatment or follow-up period for any cause, except death.

Data collection was carried out on all patients who were admitted to the emergency room for hip fractures and underwent surgery by the Orthopedic Surgery and Traumatology Service (COT).

### 2.2. Sample Size

The sample size was estimated following the procedure for finite populations, using the formula $n=\frac{N*{\left(Z_{\alpha }=1.96\right)}^{2}*p*q}{\delta^{2}*\left(N-1\right)+{\left(1.96\right)}^{2}*p*q}$. The known population reported by the National Institute of Statistics (INE) [17] and a similar study [18] was taken into account, establishing a proportion of hip fractures in the population of 0.389% (*p* = 0.000398, and its complementary *q* = 0.99602) and assuming a sampling error of 1% ($\delta^{2}$ = 0.01 in the formula). Based on this, it was concluded that the sample should be made up of 152 patients with hip fractures under care by the HUBU.

### 2.3. Main Outcomes—Instruments

The head of the Traumatology Section of the COT Service was responsible for collecting the data from each participant’s electronic medical record for further analysis.

The main study variable was cognitive impairment, assessed using PS [19]. It is a questionnaire that collects the number of errors of the evaluated patient when ten simple questions are posed, and establishes four categories of the definition of cognitive impairment depending on the dependence of people in the intellectual area: 0–2 errors is absence of deterioration or autonomy in the intellectual area, 3–4 errors is slight impairment and help of other people in intellectually complex matters, 5–7 errors is moderate deterioration and require help on a regular basis but not always, and 8–10 errors denote severe deterioration and continuous supervision. The adapted Spanish version was published in 2001 by Martínez de la Iglesia et al. [20], with an internal consistency of α = 0.84. In the present study, cognitive impairment according to PS is expressed as a dichotomous variable: absence of cognitive impairment or mild impairment (PS ≤ 4 errors) and moderate or severe cognitive impairment (PS ≥ 5 errors).

In order to study variables that may influence cognitive impairment, sociodemographic data such as age (dichotomized in <85 and ≥85 years) and gender (woman/man) and clinical data such as the type of fracture (intracapsular/extracapsular), the type of treatment (surgical/conservative), the surgical technique (arthroplasty/synthesis), complications during admission such as “delirium” or constipation, the surgical risk assessed according to the American Society of Anesthesiologists Physical Status Classification (ASA) [21], other variables that can influence cognitive impairment are institutionalization (yes/no), prescription of different drugs before admission and after hospital discharge, level of dependence using Barthel Index (BI) [22] (dichotomized in BI ≥ 60 for mild or moderate dependence and BI < 60 for severe dependence), and ambulation capacity according to the Functional Ambulation Classification (FAC) [23] (dichotomized at levels 2–5 for good or regular and 0–1 for bad) were also obtained.

Data related to clinical characteristics were collected in person or telematically at the time of admission and discharge and after six months of the episode, to study their relationship with moderate or severe cognitive impairment at the sixth month of hip fracture, as well as possible risk factors.

### 2.4. Statistical Analysis

To characterize the sample, the mean and standard deviation (SD) were used in the case of continuous variables, and absolute frequencies and percentages if the variables were categorical. Both categorical variables from more than two categories and continuous variables were dichotomized based on previous studies and tended to obtain groups as homogeneous as possible.

Bivariate analyses were performed to study the relationship between clinical features and cognitive impairment at 6 months using the Pearson independence test (χ^2^), as well as the likelihood ratio. In the analyses with significant results, the ratio of advantages or “odds” (OR) with its limits was also obtained. In addition, to quantify the magnitude of relationships of bivariate analysis and identify possible predictive factors of cognitive impairment at 6 months depending on the different clinical characteristics, an analysis was performed using binary logistic regression, adjusted for age (≥85 years) and gender (male), where all significant independent variables were included in the previous bivariate analysis. In this case, the OR with its limits was also obtained.

This analysis was carried out, taking into account the three temporal moments previously mentioned. First, bivariate and multivariate analyses studied the relationship between clinical characteristics at admission and severe moderate cognitive impairment at the sixth month of hip fracture; secondly, they were used to estimate the relationship between complications and clinical characteristics at discharge and moderate or severe cognitive impairment at the sixth month of the fracture; finally, they analyzed the relationship between clinical characteristics after the first semester of hip fracture and moderate or severe cognitive impairment at that time.

Statistical analysis was performed with SPSS software version 25 (IBM-Inc., Chicago, IL, USA). For the analysis of statistical significance, a *p*-value < 0.05 was established.

## 3. Results

The study sample consisted of 665 people, 128 of whom died during the 6 months after hip fracture. The age of the participants was between 65 and 102 years, with a mean of 86.2 years being 76.7% women (*n* = 510) and 23.3% men (*n* = 155) (Figure 1).

### 3.1. Characteristics at Admission and Cognitive Impairment at the Sixth Month in Patients with Hip Fracture

Table 1 shows the bivariate analysis of the characteristics at admission and cognitive impairment at the sixth month of the hip fracture. Moderate or severe cognitive impairment at the sixth month (PS ≥ 5) demonstrated a significant relationship with age (OR = 2.197), severe or total dependence (OR = 28.566), moderate or severe cognitive impairment (OR = 680), poor ambulation (OR = 9.474), institutionalization (OR = 10.449) and surgical risk (ASA) (OR) = 2.025) at income. The antidepressant, neuroleptic, and thickening medications that the patient had prescribed at the time of admission also demonstrated a relationship with moderate or severe impairment at the sixth month, with OR = 1.853, OR = 12.347, and OR = 5.708, respectively. However, the mental situation at the sixth month was not related to the type of fracture presented at admission.

According to the mean number of errors in the PS, cognitive impairment at admission (3.17 ± 2.81) was significantly lower than impairment at the sixth month of hip fracture (3.29 ± 3.15), with a value of *p* < 0.001. Table 2 shows an analysis by binary logistic regression adjusted for age, sex, and moderate or severe cognitive impairment at admission, taking as independent variables those significant in the previous bivariate analysis. It discovers the risk factors for the development of moderate to severe cognitive impairment at the sixth month of the fracture, severe or total dependence (OR = 15.474), and institutionalization (OR = 5.349) at admission. Age also proved to be a weak risk factor (OR = 1.078).

### 3.2. Complications during Admission and Characteristics at Discharge and Their Relationship with Cognitive Impairment at the Sixth Month in Patients with Hip Fracture

Complications during admission are generally not related to final cognitive decline at the sixth month, with the exception of “delirium” (OR = 4.110) and persistent constipation (OR = 2.243). Regarding the characteristics at hospital discharge, both severe or total dependence and poor ability to roam and institutionalization were significantly associated with moderate or severe cognitive impairment at the sixth month, with OR = 11.711, OR = 8.052 and OR = 6.627, respectively. In addition, participants with cognitive or severe impairment at the time of discharge demonstrated more than five hundred times more likely to suffer from it at the sixth month of the hip fracture than those who had no cognitive impairment or mild impairment. Other characteristics during admission and discharge such as the type of treatment or increased cognitive impairment during admission were not significant (Table 3).

Of the variables that describe the phenomena that occur during admission and discharge, adjusted for age and sex and cognitive impairment at discharge, only moderate or severe cognitive impairment and poor ability to ambulate (OR = 5.071) at discharge demonstrated a significant relationship with cognitive impairment at the sixth month in patients with hip fractures. Likewise, institutionalization at discharge was a protective factor (OR = 0.148) (Table 4).

### 3.3. Characteristics at the Sixth Month and Their Relationship with Moderate or Severe Cognitive Impairment in Patients with Hip Fracture

After analyzing the relationship between moderate or severe cognitive impairment and different characteristics and complications at the sixth month of the hip fracture, it was obtained that severe or total dependence (OR = 20.312), poor ambulation capacity (OR = 13.170), not recovering the initial gait capacity (OR = 2.377), and being institutionalized (OR = 9.491) at the sixth month, were the factors related to moderate or severe cognitive impairment in patients who suffered a hip fracture six months earlier. Having experienced a loss of at least one category in the PS at the sixth month, with respect to the initial cognitive capacity, was also significantly related to the variable and main study (OR = 9.354) (Table 5).

Taking into account the characteristics at the sixth month significantly related to moderate or severe cognitive impairment, Table 5 shows a multivariate study that confirms as risk factors for cognitive impairment severe or total dependence at the sixth month (OR = 8.088), cognitive impairment of a category of PD during the semester (OR = 7.024), and institutionalization during the sixth month from the time of fracture (OR = 6.317) (Table 6).

## 4. Discussion

The results of this research show that certain clinical factors of patients with hip fractures are related to moderate or severe cognitive impairment at the sixth month of the fracture. In general, age, initial moderate or severe cognitive impairment at discharge, cognitive worsening at the sixth month with respect to the time of admission, severe or total dependence on admission and six months after the fracture, poor ambulation at discharge, and institutionalization prior to admission or during the first semester after the hip fracture were the main factors related to moderate or severe cognitive impairment at the sixth month.

Cognitive impairment prior to the hip fracture or occurring during admission is a risk factor that compromises independence in performing activities of daily living in patients with hip fractures, at least during the two and twelve months thereafter [24,25].

The data show, through multifactorial analysis with multiple linear regression, that the mental state at the sixth month is fundamentally related to the previous mental state, additional conditioning factors, the non-recovery of independence (BI), the institutionalization at the sixth month, and the “bad” gait after hospital discharge.

Holmes et al. [2] reviewed in their study the prevalence and effect of dementia, depression, and delirium in the elderly population suffering from hip fracture, and subsequent follow-up demonstrated increased dependence and mortality in such patients. Likewise, more recent studies report the effect of cognitive impairment on poor functional evolution [26,27,28], although Givens et al. [27] consider that these limitations do not affect hip fracture surgeries beyond the sixth month.

In their study to assess the association between dementia and postoperative outcomes of older adults with hip fractures, Seitz et al. [29] found that dementia is associated with a poor prognosis after hip fracture surgery. These results are in line with those obtained in the present research since an older age in people with hip fractures has been significantly related to moderate or severe cognitive impairment at the sixth month of the episode. In this investigation, age has, after multivariate adjustment, a weak association, with OR = 1.078 (1.002–1.160), with respect to cognitive deterioration.

Moderate to severe cognitive impairment at discharge persists six months after the hip fracture, results that coincide with those obtained by Peeters et al. [30] and Alexiou et al. [31], who also found that cognitive impairment occurs more in women with previous pathologies, long hospital stays, and complications during admission, and even more so in patients undergoing osteosynthesis compared to those operated by hip arthroplasty, whose cognitive evolution is better. A worse cognitive level was found at the sixth month after the hip fracture, unrelated to any other comorbidity, to the hospital stay, or to the type of treatment or surgical technique, or other complications during admission, except for “delirium”. Likewise, a worse cognitive capacity at the fourth month of the fracture has been related to a worse evolution of the ability to perform the activities of daily life [32] and with the presence of cognitive impairment, measured with PS, at admission [33]. As in the previously mentioned studies, our results show a significant relationship between cognitive impairment at the sixth month and moderate or severe cognitive impairment and poor ability to ambulate at discharge, and severe or total dependence during the first semester after hip fracture. Regarding the level of dependence, Córcoles-Jiménez et al. [34] found in their prospective study a significant relationship between a lower recovery of functional independence one year after hip fracture, measured with IB, and a greater number of errors in PS at admission.

Taking into account the results of our research, institutionalization during the first semester after the hip fracture is also related to moderate to severe cognitive impairment in the medium term. On the other hand, institutionalization has been shown to reduce mortality from hip fractures according to the results obtained by Seitz et al. [29] in their study, which associates this fact with an average survival of three more years in patients with hip fractures. Institutionalization at hospital discharge, and at the sixth month, have been found to be protective factors for cognitive deterioration in the medium term since they are patients who previously have a better functional and mental status since they live at home.

As for its strengths, it is an investigation with a large sample that includes all the hip fractures treated at the HUBU. In addition, it considers a wide variety of clinical factors relevant to its subsequent bivariate and multivariate analysis, which allows the elimination of interaction biases and draws conclusions with greater evidence. In addition, despite following a retrospective methodology, it is a longitudinal study, which to a large extent allows us to assimilate the conclusions to an etiopathogenic explanation of cognitive impairment in the elderly hip-fractured patients in our environment.

The results of this study should be considered in the context of its limitations. The assessment of the cognitive status of the patients was not based on diagnosis of dementia or “major cognitive disorder” or “mild cognitive disorder” according to criteria of the American Psychiatric Association (APA) [6]. However, PS is also widely used in studies on elderly patients with hip fractures, with validity demonstrated in the Spanish version [22], and even equivalence with the diagnosis of dementia for some authors [35]. It is also a relative limitation that the six-month time perspective is lower than many publications that take a year or more. It should also be noted that the scientific evidence of this study is less than that of a cohort study, mainly because it is a mere consultation of recorded data, no matter how rigorous the anamnesis and registration of it has been.

## 5. Conclusions

This research provides evidence about the clinical factors that predict moderate or severe cognitive impairment at the sixth month in patients undergoing surgery for a hip fracture. These factors are age over 85 years; moderate or severe cognitive impairment at initial admission and discharge, as well as cognitive worsening at the sixth month with respect to the time of admission; severe or total functional dependence at admission and six months after the fracture; poor ambulation at discharge; and institutionalization prior to hospital admission or during the first semester. Prospective studies over an extended period of time are suggested.

## Figures and Tables

**Figure 1 jcm-11-02608-f001:**
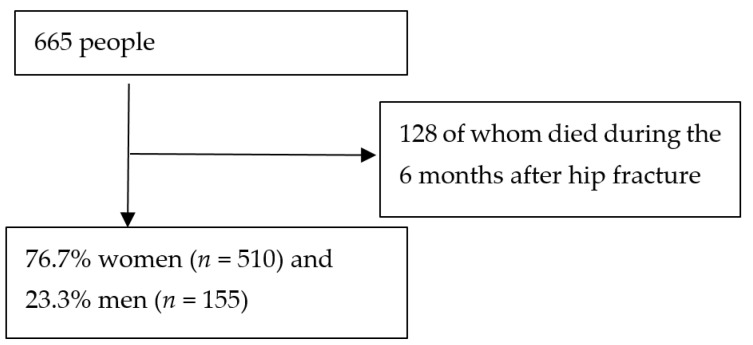
Participants flow.

**Table 1 jcm-11-02608-t001:** Chi^2^ test results between clinical features at admission and cognitive impairment at sixth month.

Characteristics at the Entrance	PS 6th Month		Chi^2^ Test		OR		
	PS ≤ 4	PS ≥ 5	χ^2^ χ RV^2^	*p*-Value	Rho	Lower Limit	Upper Limit
**Age**							
<85 years (*n* = 208)	171 (82.2%)	37 (17.8%)	12.85	<0.001 <0.001	0.455	0.298	0.696
≥85 years (*n* = 329)	223 (67.8%)	106 (32.2%)	14.11		2.197	1.438	3.357
**Type of fracture**							
Intracapsular (*n* = 225)	170 (43.1%)	55 (38.5%)	0.76	0.382	*p* > 0.05		
Extracapsular (*n* = 312)	224 (56.9%)	88 (61.5%)	0.95	0.329			
**Antidepressants**							
Yes (*n* = 148)	95 (64.2%)	53 (35.8%)	8.18	0.004	1.853	1.230	2.794
No (*n* = 398)	299 (76.9%)	90 (23.1%)	8.5	0.004	0.54	0.358	0.813
**Neuroleptics**							
Yes (*n* = 85)	23 (27.1%)	62 (72.9%)	108.06	<0.001	12.347	7.228	21.092
No (*n* = 482)	371 (82.1%)	81 (17.9%)	98.12	<0.001	0.081	0.047	0.138
**Thickeners**							
Yes (*n* = 9)	3 (33.3%)	6 (66.7%)	5.57	0.018	5.708	1.408	23.136
No (*n* = 528)	391 (74.1%)	137 (25.9%)	6.41	0.011	0.175	0.043	0.71
**BI**							
≥60 (*n* = 552)	476 (95.2%)	76 (46.1)	143.71	<0.001	0.035	0.017	0.071
<60 * (n* = 114)	25 (4.8%)	89 (53.9)	131.12	<0.001	28.566	14.046	58.097
**PS**							
≤4 (*n* = 414)	391 (94.4%)	23 (5.6%)	406.15	<0.001	0.001	0.000	0.005
≥5 (*n* = 123)	3 (2.4%)	120 (97.6%)	416.56	<0.001	680.0	200.7	2304.2
**Ability to roam**							
Good or regular (*n* = 486)	380 (78.2%)	106 (21.8%)	58.25	<0.001	0.106	0.055	0.203
Bad (*n* = 51)	14 (27.5%)	37 (72.5%)	52.66	<0.001	9.474	4.938	18.177
**Previous institutionalization**							
No (*n* = 391)	338 (86.4%)	53 (13.6%)	123.37	<0.001	0.098	0.063	0.152
Yes (*n* = 146)	56 (38.4%)	90 (61.6%)	117.72	<0.001	10.249	6.590	15.942
**Surgical risk**							
ASA I+II (*n* = 265)	213 (54.8%)	52 (37.4%)	11.64	<0.001	0.494	0.332	0.735
ASA III+IV (*n* = 263)	176 (45.2%)	87 (62.6%)	12.43	<0.001	2.025	1.361	3.013

PS: Pfeiffer Scale; OR: Odds ratio; BI: Barthel Index; ASA: American Society of Anesthesiologists Physical Status Classification.

**Table 2 jcm-11-02608-t002:** Results of binary logistic regression to estimate the relationship between clinical characteristics at admission and cognitive impairment at the sixth month.

Characteristics at the Entrance	LR		OR		
	χ ^2^ Wald	*p*-Value	Rho	Lower Limit	Upper Limit
Age	4.053	0.044	1.078	1.002	1.160
Gender: male	0.289	0.591	0.733	0.236	2.277
PS: ≥ 5	71.168	<0.001	535.762	124.437	2306.709
Antidepressants: Yes	0.106	0.744	0.844	0.305	2.338
Neuroleptics: Yes	0.457	0.499	1.620	0.400	6.567
Thickeners: Yes	0.021	0.884	0.778	0.027	22.350
BI: < 60	10.802	0.001	15.474	3.021	79.249
Ability to roam: Poor	2.116	0.146	0.221	0.029	1.689
Previous institutionalization: Yes	12.236	<0.001	5.349	2.090	13.687
Surgical risk: ASA III or IV	0.148	0.700	1.205	0.467	3.108

LR: Binary logistic regression; OR: Odds ratio; PS: Pfeiffer Scale; BI: Barthel Index; ASA: American Society of Anesthesiologists Physical Status Classification.

**Table 3 jcm-11-02608-t003:** Results of the Chi^2^ test among complications and characteristics at discharge and their relationship with cognitive impairment at the sixth month.

Complications During Admission and Characteristics at Discharge	PS 6th Month		Chi^2^ Test		OR		
	PS ≤ 4	PS ≥ 5	χ^2^ χ RV ^2^	*p*-Value	Rho	Lower Limit	Upper Limit
**Type of treatment**							
Surgical (*n* = 528)	389 (73.7%)	139 (26.3%)	0.7 1.34	0.401 0.247	*p* > 0.05		
Conservative (*n* = 9)	5 (55.6%)	4 (44.4%)					
**Surgical intervention**							
Synthesis (*n* = 321)	234 (72.9%)	87 (27.1%)	0.16 0.26	0.686	*p* > 0.05		
Arthroplasty (*n* = 207)	155 (74.9%)	52 (25.1%)		0.613			
**“Delirium”**							
Yes (*n* = 170)	91 (23.1%)	79 (55.2%)	48.64 47.93	<0.001 <0.001	4.110	2.743	6.159
No (*n* = 367)	303 (76.9%)	64 (44.8%)			0.243	0.162	0.365
**Constipation**							
Yes (*n* = 226)	145 (36.8%)	81 (56.6%)	16.14 16.8	<0.001 <0.001	2.243	1.521	3.310
No (*n* = 311)	249 (63.2%)	62 (43.4%)			0.446	0.302	0.658
**IB upon discharge**							
≥60 (*n* = 390)	340 (87.2%)	50 (12.8%)	136.48 130.40	<0.001 <0.001	0.085	0.055	0.134
<60 (*n* = 147)	54 (36.7%)	93 (63.3%)			11.711	7.483	18.327
**Worsening BI during admission**							
No (*n* = 195)	110 (56.4%)	85 (43.6%)	43.73 43.93	<0.001 <0.001	3.784	2.537	5.644
Yes (*n* = 342)	284 (83%)	58 (17%)			0.264	0.177	0.394
**PS at discharge**							
≤4 (*n* = 411)	390 (94.9%)	21 (5.1%)	410.51 421.13	<0.001 <0.001	0.002	0.001	0.005
≥5 (*n* = 126)	4 (3.2%)	122 (96.8%)			566.429	190.746	1682.031
**Worsening PS during admission**							
No (*n* = 518)	380 (73.4%)	138 (26.6%)	0.00 0.00	>0.05	*p* > 0.05		
Yes (*n* = 19)	14 (73.7%)	5 (26.3%)		>0.05			
**Ability to roam**							
Good or regular (*n* = 342)	301 (88%)	41 (12%)	101.28 101.69	<0.001 <0.001	0.124	0.081	0.191
Bad (*n* = 195)	93 (47.7%)	102 (52.3%)			8.052	5.235	12.385
**Institutionalization**							
No (*n* = 229)	272 (88.3%)	36 (11.7%)	80.74 83.77	<0.001 <0.001	0.151	0.098	0.233
Yes (*n* = 308)	122 (53.3%)	107 (46.7%)			6.627	4.294	10.226

PS: Pfeiffer Scale; OR: Odds ratio; BI: Barthel Index; ASA: American Society of Anesthesiologists Physical Status Classification.

**Table 4 jcm-11-02608-t004:** Results of binary logistic regression to estimate the relationship between complications during admission and characteristics at discharge and cognitive impairment at the sixth month.

Complications During Admission and Characteristics at Discharge	LR		OR		
	χ ^2^ Wald	*p*-Value	Rho	Lower Limit	Upper Limit
Age	1.388	0.239	1.046	0.97	1.128
Gender: male	0.176	0.675	0.794	0.27	2.334
PS at discharge: ≥ 5	82.215	<0.001	547.91	140.19	2141.45
“Delirium”: Yes	1.697	0.193	0.510	0.185	1.405
Persistent constipation: Yes	0.218	0.640	1.245	0.496	3.124
BI at discharge: < 60	0.194	0.660	1.266	0.443	3.619
IB worsening during admission: Yes	0.027	0.869	1.174	0.174	7.938
Ability to roam: Poor	9.105	0.003	5.071	1.766	14.556
Institutionalization at discharge: Yes	13.045	<0.001	0.148	0.053	0.418

LR: Binary logistic regression; OR: Odds ratio; PS: Pfeiffer Scale; BI: Barthel Index.

**Table 5 jcm-11-02608-t005:** Results of the Chi^2^ test between the characteristics at the sixth month and its relationship with cognitive impairment.

Characteristics of the Sixth Month	EP 6th Month		Chi^2^ Test		OR		
	PS ≤ 4	PS ≥ 5	χ^2^ χ RV ^2^	*p*-Value	Rho	Lower Limit	Upper Limit
**BI**							
≥60 (*n* = 405)	358 (88.4%)	47 (11.6%)	187.24 176.96	<0.001 <0.001	0.049	0.030	0.080
<60 (*n* = 132)	36 (27.3%)	96 (72.7%)			20.312	12.455	33.125
**Ability to roam**							
Good or regular (*n* = 402)	349 (86.8%)	53 (13.2%)	145.23	<0.001 <0.001	0.076	0.048	0.120
Bad (*n* = 135)	45 (33.3%)	90 (66.7%)	137.11		13.170	8.314	20.861
**Recovery of initial gait capacity**							
Yes (*n* = 298)	241 (80.87%)	57 (19.13%)	18.43	<0.001	0.421	0.285	0.622
No (*n* = 239)	153 (64.02%)	86 (35.98%)	19.25	<0.001	2.377	1.607	3.515
**Institutionalization**							
No (*n* = 303)	275 (90.8%)	28 (9.2%)	105.57	<0.001 <0.001	0.105	0.066	0.168
Yes (*n* = 234)	119 (50.9%)	115 (49.1%)	111.40		9.491	5.958	15.120
**Worsening PS since admission**							
No (*n* = 459)	370 (80.6%)	89 (19.4%)	82.23	<0.001 <0.001	0.107	0.063	0.182
Yes (*n* = 78)	24 (30.8%)	54 (69.2%)	74.64		9.354	5.486	15.949

PS: Pfeiffer Scale; OR: Odds ratio; BI: Barthel Index.

**Table 6 jcm-11-02608-t006:** Results of binary logistic regression to estimate the relationship between characteristics at the sixth month and their relationship with cognitive impairment.

Characteristics of the Sixth Month	LR		OR		
	χ ^2^ Wald	*p*-Value	Rho	Lower Limit	Upper Limit
Age	0.523	0.470	0.985	0.946	1.026
Gender: Male	0.002	0.961	1.016	0.531	1.944
BI: < 60	29.881	<0.001	8.088	3.823	17.115
Ability to roam: Poor	2.732	0.098	1.884	0.889	3.993
Recovery of initial gait capacity: No	1.395	0.238	1.418	0.794	2.529
Institutionalization: Yes	33.357	<0.001	6.317	3.379	11.807
Worsening PS since admission: Yes	27.627	<0.001	7.024	3.396	14.531

LR: Binary logistic regression; OR: Odds ratio; PS: Pfeiffer Scale; BI: Barthel Index.

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
