# Peer review of "Predictors of Moderate or Severe Cognitive Impairment at Six Months of the Hip Fracture in the Surgical Patient over 65 Years of Age"

_jcm, 2022, doi:10.3390/jcm11092608_

Round 1

Reviewer 1 Report

Thank you for the opportunity to review this manuscript. The purpose of the study aims to determine the relationship 70 between significant cognitive impairment and the different clinical characteristics of 71 patients aged 65 years or older with hip fracture, during the six months after that episode.

I completely see the interest of the topic for the readership of the scope of medicina, however, I’ve some comments that need to be addressed before considering the paper for publication.

Abstract:

The Abstract is generally well written.

Manuscript:

Introduction:

Generally, the introduction covers all necessary content deduce the purpose of the study. However, there are still some modifications that you should consider to revise.

Authors should explain better in the introduction the Moderate or Severe Cognitive Impairment the correlations with Hip Fracture in the Surgical Patient (needs specific reference). This will give a strong background to their research.

Please provide a research hypothesis according to your purpose.

Materials and Methods

Please Well elaborated section on the sample recruitment criteria.

Please add information about the research procedure.

Please provide reference for each outcome measure that you use and their specific purpose.

Results:

According to CONSORT statement, the participants flow should be placed in the results section.

Discussion:

Discussion is not meant to just repeat a result and give any literature references that say the similar. In discussing your findings, you should give us logic interpretation of what the findings mean, in general, in specific and how that is linked or underlined by findings of literature. You should not only list your results again and list other results that fit kind of arbitrarily.

Please give us your main finding in the first section of the Discussion.

Describe the limitations of the study at the end of the discussion.

Author Response

Prof. Jerónimo J. González-Bernal, PhD

Department of Health Sciences

University of Burgos. Paseo Comendadores, s/n.

Burgos, 09001, Spain

20-04-2022

JCM.Subject: Submissions Needing Revision

Dear  Ms. Andreea Strugariu Assistant Editor

Thank you very much for inviting us to submit a revised version of our manuscript (JCM-1650949) entitled:Predictors of Moderate or Severe Cognitive Impairment at Six Months of the Hip Fracture in the Surgical Patient Over 65 Years of Age”

We have checked our manuscript according to the Academic Editor, the reviewer comments and the Journal requirements. (We have also responded to some comments from reviewers point by point).

We would be very grateful if you could consider our manuscript to be published in your journal.

  • We have included a revised manuscript file v.2

Yours sincerely,

Prof. Jerónimo J. González-Bernal

Reviewer 2 Report

Minor concernings to improve the overall quality:

  • A cross-sectional study determined the cognitive impairment level influence on descriptive characteristics, comorbidities, complications, and pharmacological features of older adults with hip fractures. Significant differences (p < .05, R = .012-.475) between cognitive impairment levels were shown. Shorter presurgery hospital length of stay and lower depression and Parkinson comorbidities; delirium complication; and antidepressants, antiparkinsonians, and neuroleptics use were shown for the no-impairment group. please cite doi:10.1097/rnj.0000000000000159
  • Hip fractures in persons with cognitive impairments represent a major public health issue in older populations that often results in poor health-related quality of life (HRQoL). Nursing care interventions for older persons with cognitive impairment should be initiated immediately after surgery for hip fracture to prevent a significant decline in HRQoL. The influences of hip fracture type and surgical approach on changes in HRQoL suggest a need for further investigations to determine what contributed to the observed inconsistencies in the outcomes. cite doi:10.1097/jnr.0000000000000371

Author Response

(The authors gave the same response as above.)
